# Pix2Scene: Learning Implicit 3D Representations from a Single Image

## Abstract

We aim to model 3D properties of the scenes from single 2D images. Learning 3D scenes from 2D images is a long-standing problem in computer vision with applications in related fields such as simulation and robotics. We propose **pix2scene**, a deep generative-based approach that represents the 3D scene in a learnt latent variable decoded into a viewpoint-dependent representation that can be rendered. Our method learns the depth of the scene and leverages a local smoothness assumption to extract the orientation of visible scene points. We achieve this using an encoder-decoder adversarial learning mechanism and a novel differentiable renderer to train the 3D model in an end-to-end fashion, using only images. We showcase the generative ability of our model qualitatively on the ShapeNet dataset (Chang et al., 2015). We also demonstrate that our model can predict the structure of scenes from various, previously unseen view points. Finally, we evaluate the effectiveness of the learned 3D scene representation in supporting 3D spatial reasoning.

## 1 Introduction

Understanding the 3-dimensional (3D) world from its 2-dimensional (2D) projections is a fundamental problem in computer vision with a broad range of application in robotics, simulation and design. Given that the majority natural scene data is available exclusively in the form of 2D images, the ability to directly infer knowledge about 3D structure from these images would be of great utility in scene understanding.

Inferring the 3D structure from multiple images of a scene has been pursued extensively, such as in stereo or structure from motion tasks (Hartley & Zisserman, 2004). Since most available natural image data informative about the real world comes with only a single view of a given scene, it is perhaps more important to explore the development of models which can infer the 3D structural properties from *a single image*. On the other hand, single image 3D recovery is an extremely challenging and heavily under constrained task. The system has to rely on prior knowledge and 2D visual cues such as textures, shadows or occlusions in order to provide hints to the 3D structure of the scene. Practically, building a machine learning model that learns to infer 3D structure from images requires either a strong inductive bias or supervision. While some have used the 3D ground truth as explicit supervision (Wu et al., 2016; 2015), in most cases of interest, such supervision will not be available. Consequently, our long term goal is to *infer the 3D structure of realistic scenes from single images*. In this paper we take a step towards this direction via a method of unsupervised learning of the 3D structure, directly from a single 2D image of each scene. Our method based on the adversarial learning framework (Goodfellow et al., 2014) and exploits a uniquely suitable 3D representation (i.e., surfels (Pfister et al., 2000)) and a differentiable renderer.

Most 3D reconstruction methods rely on representing 3D objects *explicitly* using either voxels (Rezende et al., 2016; Yan et al., 2016) or meshes (Kanazawa et al., 2018; Wang et al., 2018). *Explicit* representations store all the rendering-relevant information from a given 3D space and are easily transferable, i.e., they can be loaded with any 3D modeling software and viewed from any angle. However, approaches using explicit representations typically scale very poorly ($O(n^3)$) or require a sparse/discrete representation which can be challenging for deep learning methods. As a result, these representations have only been applied to the reconstruction of single objects. As an alternative we propose to learn an *implicit 3D representation* which produces only the 3D geometry which is directly relevant for a particular viewpoint. Our viewpoint-specific 3D geometry is captured

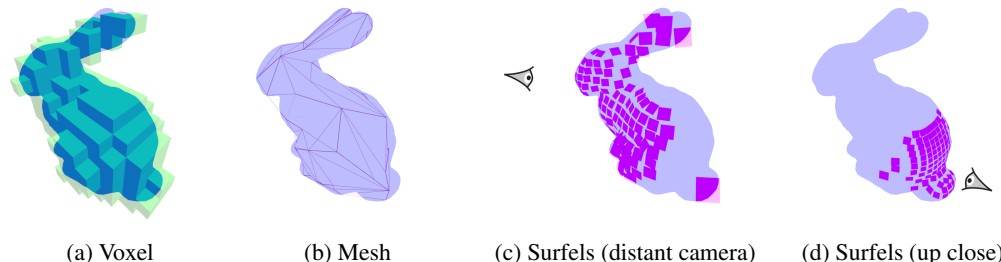

| (a) Voxel | (b) Mesh | (c) Surfels (distant camera) | (d) Surfels (up close) |

Figure 1: **Implicit vs explicit representations.** Explicit voxel and mesh representations are viewpoint-independent and constitutes the complete scene. Our implicit surfel-based representation is viewpoint-dependent and it adapts the resolution to the viewpoint. The full scene is contained in a high-dimensional latent variable and only when the scene is to be rendered, the latent variable is serialized to surfels for a specific view.

using camera facing *surfels* (Pfister et al., 2000) which are surface elements defined by its position, orientation and material properties. Given an image we can infer its implicit 3D representation and then recreate novel surfel representations of the underlying scene from unobserved viewpoints. In general, we note that in a 3D scene, only a small fraction of the entities are perceivable from the camera. As the camera moves, and the occluded regions become visible, our method then generates surfels for those newly unoccluded regions. Another advantage of this approach is that minimal number of primitives (surfels) are required to obtain a high-resolution image as the camera moves closer to a part of the scene. Moreover this representation fits well with image based convolutional architectures.

Our model, *Pix2Scene*, is a deep generative-based approach for modelling the 3D structure of a scene directly from images. This model is unsupervised in the sense that it does not require 3D groundtruth or any other kind of image annotations. We base our model on Adversarially Learned Inference (ALI) approach (Dumoulin et al., 2016). ALI extends the GAN (Goodfellow et al., 2014) framework by learning to infer the latent representation of a given image. In pix2scene the learned latent space embeds the 3D information of the underlying scene. The latent representation is mapped via a decoder network to a view-dependent 3D surface and then projected to image space by a differentiable renderer. The resulting image is then evaluated by an adversarial critic.

While our long-term goal is to be able to infer the 3D structure of a real-world photograph, in this paper we experiment exclusively with synthetically-constructed scenes and adopt several simplifying assumptions. In particular, we assume that the world is piece-wise smooth and that for each input image the illumination, view and object materials are known.

This work has the following main contributions, (1) we propose a novel unsupervised method for 3D understanding from a single image; (2) we propose a new implicit 3D representation based on *view-space* surfels; (3) we propose a surfel-based differentiable 3D renderer that can be used as a layer of a neural network; and (4) we propose 3D-IQTT a new 3D understanding evaluation benchmark. This task evaluates the model's ability to perform mental rotation by obtaining comprehensive understanding of underlying 3D structure. We also estimate the camera pose as part of the learnt latent variable for this particular task.

## 2 RELATED WORK

The generation and reconstruction of 3D objects from images has been studied extensively in the computer vision and graphics communities (Saxena et al., 2009; Chaudhuri et al., 2011; Kalogerakis et al., 2012; Chang et al., 2015; Rezende et al., 2016; Soltani et al., 2017). Our work bears some conceptual similarities to Kulkarni et al. (2015) which casts the 3D reconstruction problem as a more traditional inverse graphics task. By using Variational Auto-Encoder(VAEs) (Kingma & Welling, 2014), they learn a representation of objects that disentangles factors of variations from images (i.e., object pose and configuration) and use the approach for specific transformations such as out of axis rotation. However, unlike their approach, ours is fully unsupervised and we implicitly generate

3D structure of scenes from single images. Our mechanism learns a latent representation for the underlying scene, which can later be used to render from different views and lighting conditions. Similar to ours, Rezende et al. (2016) infer the 3D configuration at their output. They adopt a probabilistic inference framework to build a generative model for 3D by combining a standard projection mechanism with gradient estimation methods. In particular, their approach requires multiple runs with mechanisms such as REINFORCE (Williams, 1992) in order to infer gradient from the projection layer. In addition, their use of mesh and voxel representations could become an obstacle to scaling their method to more complex scenes. Our approach is not susceptible to restrictions imposed by meshes or other scaling issues and has the potential to adapt to arbitrary scene configurations.

## 3 METHOD

### 3.1 IMPLICIT 3D REPRESENTATION AND SURFELS

Explicitly representing 3D structure presents different challenges for generative models (Kobbelt & Botsch, 2004). Representing entire objects using voxels scales poorly given its ($O(n^3)$) complexity. The vast majority of the generated voxels aren't relevant to most viewpoints, such as the voxels that are entirely inside of objects. A common workaround is to use a sparse representation such as meshes. However, these too come with their own drawbacks, such as the need to discretise complex objects. This makes mesh representation difficult to generate using neural networks. Current mesh based methods mainly rely on deforming a pre-existing mesh.

On the other hand, our implicit approach represents the 3D scene in a high-dimensional latent variable. In our framework, this latent variable (or vector) is decoded using a generator network into a viewpoint-dependent representation of surface elements — similar to the surfels (Pfister et al., 2000) — that constitute the visible part of the scene. This representation is very compact: given a renderer's point of view, we can represent only the part of the 3D surface needed by the renderer. Also, as the camera moves closer to a part of the scene, our generator will allocate more surfels to represent that part of the scene and thereby increasing the resolution. Figure 1 illustrates these different representations. For descriptive purpose, surfels are shown as squares, but in general they do not have any shape.

Formally, surfels are represented as a tuple $(P, N, \rho)$, where $P = (p_x, p_y, p_z)$ is its 3D position, $N = (n_x, n_y, n_z)$ is the surface normal vector, and $\rho = (k_r, k_g, k_b)$ is the reflectance of the surface material. Since we are interested in modelling structural properties of the scenes i.e., geometry and depth, we assume that objects in the scene have a uniform material. We represent the surfels in the camera coordinate system. This significantly reduces the number of surfels by considering only the ones that will get projected onto a pixel in the rendered image. Moreover, this allows to reduce the position parameters to only $p_z$ being this the distance along a ray going through the surfel to the center of its pixel.

### 3.2 DIFFERENTIABLE 3D RENDERER

As the critic operates only on image space, we need to project the generated 3D representations back to the 2D space using a renderer. In our setting, each stage of the rendering pipeline must be differentiable to allow us to take advantage of gradient-based optimization and backpropagate the critic's error signal to the surfel representation. The rendering process can be partitioned into two stages. During forward-propagation, the first stage finds the mapping between the surfels and the pixels; and the second stage computes the color of each pixel. During back-propagation, the first stage directs the gradients only to the surfels that get projected onto the image; and the second stage is differentiable as long as the shading operations are differentiable.

The first stage of the rendering involves finding the mapping between the surfels and the pixels. This requires performing the expensive operation of ray object intersection (See Figure 2a). Our model requires a fast rendering engine as it will be use in every learning iteration. Conventional ray tracing algorithms are optimized for generating multiple views from the same scene. However in our setting during learning we render only one image from each scene. Moreover ray tracing algorithms require from representing the full scene, which is very inefficient as we only represent the part visible by the camera. To resolve these issues, our generator proposes one surfel for each pixel in the camera's

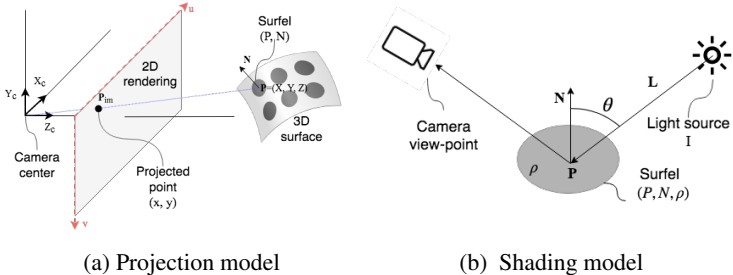

(a) Projection model         (b) Shading model

Figure 2: **Differentiable 3D renderer.** (a) A surfel is defined by its position $P$, normal $N$, and reflectance $\rho$. Each surfel maps to an image pixel $P_{im}$. (b) The surfel's color depends on its reflectance $\rho$ and the angles $\theta$ between each light $I$ and the surfel's normal $N$.

coordinate system. Our PyTorch implementation of the differentiable renderer can render a $128 \times 128$ surfel-based scene in under 1.4 ms on a mobile NVIDIA GTX 1060 GPU.

The color of a surfel depends on the material reflectance, its position and orientation, and the ambient and point light source colors. See Figure 2b. Given a surface point $P_i$, the color of its corresponding pixel $I_{rc}$ is given by the shading equation:

$$I_{rc} = \rho_i(L_a + \sum_j \frac{1}{k_l\|d_{ij}\| + k_q\|d_{ij}\|^2} L_j \max\left(0, N_i^T d_{ij}/\|d_{ij}\|\right)), \tag{1}$$

where $\rho_i$ is the surface reflectance, $L_a$ is the ambient light's color, $L_j$ is the $j^{\text{th}}$ positional light source's color, with $d_{ij} = L_j^{\text{pos}} - P_i$, or the direction vector from the scene point to the point light source, and $k_l$, $k_q$ being the linear and quadratic attenuation terms respectively. Equation 1 is an approximation of rendering equation proposed in Kajiya (1986).

### 3.3 PIX2SCENE MODEL

The adversarial training paradigm allows the generator network to capture the underlying target distribution by competing with an adversarial critic network. Pix2scene employs bi-directional adversarial training to model the distribution of surfels from just 2D images.

#### 3.3.1 BI-DIRECTIONAL ADVERSARIAL TRAINING

ALI (Dumoulin et al., 2016) or Bi-GAN (Donahue et al., 2016) extends the GAN (Goodfellow et al., 2014) framework by including the learning of an inference mechanism. Specifically, in addition to the decoder network $G_x$, ALI provides an encoder $G_z$ which maps data points $x$ to latent representations $z$. In these bi-directional models, the critic, $D$, discriminates in both the data space ($x$ versus $G_x(z)$), and latent space ($z$ versus $G_z(x)$), maximizing the adversarial value function over two joint distributions. The final min-max objective can be written as:

$$\min_G \max_D \mathcal{L}_{ALI}(G, D) := \mathbb{E}_{q(x)}[\log(D(x, G_z(x)))] + \mathbb{E}_{p(z)}[\log(1 - D(G_x(z), z))], \tag{2}$$

where $q(x)$ and $p(z)$ denote encoder and decoder marginal distributions.

#### 3.3.2 MODELLING DEPTH AND CONSTRAINED NORMAL ESTIMATION

Based on the ALI formulation, as depicted in Figure 3, our model has an encoder network which captures the distribution over the latent space given an image data point $x$. The decoder network maps a fixed latent distribution $p(z)$ (a standard normal distribution in our case) to the 3D surfel representation. Next, the surfel representation are rendered into a 2D image using our differentiable renderer. The resulting image is then given as input to the critic to distinguish from the real image data. Note that the input to the critic comes from the joint space of data with its corresponding latent code, as in ALI.

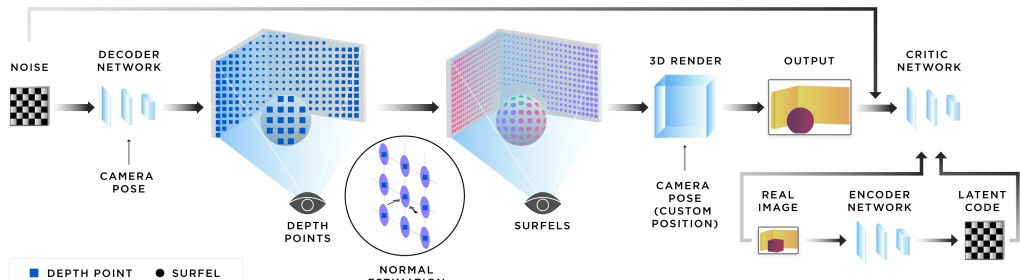

Figure 3: **Pix2scene model.** Pix2scene generates realistic 3D views of scenes by training on 2D images alone. Its decoder generates the surfels depth $p_z$ from a noise vector $z$ conditioned on the camera pose. The surfels normal is estimated from its predicted depth. The surfels are then rendered into a 2D image and together with image samples from the target distribution are fed to the critic.

A straightforward way to model the decoder network could be to learn a conditional distribution to produce the surfel's depth ($p_z$) and normal ($N$). But this could lead to inconsistencies between the local shape and the surface normal. For instance, the decoder can fake an RGB image of a 3D shape simply by changing the normals while keeping the depth fixed. To avoid this issue, we exploit the fact that real-world surfaces are locally planar, and surfaces visible to the camera have normals constrained to be in the half-space of visible normal directions from the camera's view point. Considering the camera to be looking along the $-z$ direction, the estimated normal has the constraint $n_z > 0$. Therefore, the local surface normal is estimated by solving the following problem for every surfel,

$$\|N^T \nabla P\| = 0$$
$$\text{subject to,} \|N\| = 1 \text{ and } n_z > 0, \tag{3}$$

where the spatial gradient $\nabla P$ is computed using 8 neighbour points, and $P$ is the position of the surfels in the camera coordinate system obtained by backprojecting the generated depth along rays.

This approach enforces consistency between the predicted depth field and the computed normals. If the depth is incorrect, the normal-estimator outputs an incorrect set of normals, and result in an RGB image inconsistent with the data distribution, which would in-turn get penalized by the critic. The decoder and the encoder networks are thus incentivized to predict realistic depths.

### 3.3.3 MODEL TRAINING

The Wasserstein-GAN (Arjovsky et al., 2017) formalism provides stable training dynamics using the 1-Wasserstien distance between the distributions. We adopt the gradient penalty setup as proposed in Gulrajani et al. (2017) for more robust training, however, we modify the formulation to take into account the bidirectional training.

Architectures of our networks, and training hyper parameters are explained in detail in appendix A. Breifly, we used Conditional Normalization (Dumoulin et al., 2016; Perez et al., 2017) for conditioning the view point (or camera pose) in the encoder, decoder and the discriminator networks. The view point is a three dimensional vector representing positional co-ordinates of the camera. In our training, the affine parameters of the Batch-Normalization layers (Ioffe & Szegedy, 2015) are replaced by learned representations based on the view point. The final objective includes a bi-directional reconstruction loss as formulated in Equation 4. This in-turn enforces the reconstructions from the model to stay close to the corresponding inputs. We also use a reconstruction error in the objective function for the encoder and decoder networks as it has been empirically shown to improve reconstructions in ALI-type models Li et al. (2017).

$$\mathcal{L}_{recon} = \mathbb{E}_{q(\boldsymbol{x})}[||\boldsymbol{x} - rend(G_x(G_z(\boldsymbol{x})))||_2] + \mathbb{E}_{p(\boldsymbol{z})}[||\boldsymbol{z} - G_z(rend(G_x(\boldsymbol{z})))||_2] \tag{4}$$

where function $rend(\cdot)$ denotes rendered image on the decoder side.

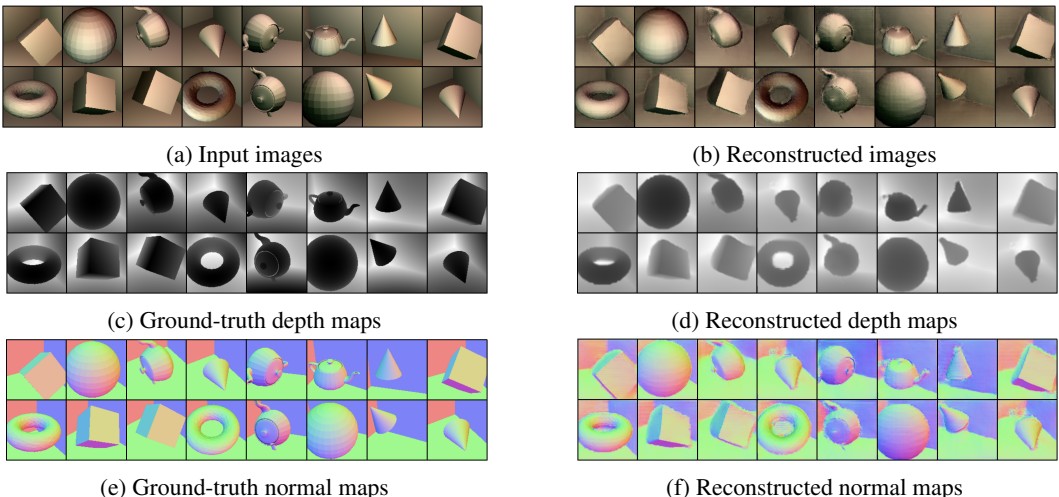

(a) Input images

(b) Reconstructed images

(c) Ground-truth depth maps

(d) Reconstructed depth maps

(e) Ground-truth normal maps

(f) Reconstructed normal maps

Figure 4: **Scene reconstruction. Left:** Input images of rotated objects into a room with its depth and normal groundtruth maps. **Right:** pix2scene reconstructions with its depth and normal maps.

|  | **Box scenes** | | **Shape scenes** |
|---|---|---|---|
|  | rand Tr. | rand Rot. | rand Rot. |
| Hausdorff-F | 0.087 | 0.102 | 0.125 |
| Hausdorff-R | 0.093 | 0.183 | 0.191 |
| MSE-depth | 0.032 | 0.022 | 0.038 |

Table 1: **Scene reconstruction results.** Hausdorff metric on 3D surfels and MSE on the depth maps.

## 4 EXPERIMENTS

### 4.1 EXPERIMENTAL SETUP

**Tasks.**   Our model is capable of both reconstructing 3D scenes and generate new ones. We evaluate the 3D understanding capability of the model on 3D-IQTT: a spatial reasoning based semi-supervised classification task. The goal of the 3D-IQTT is to quantify ability of our model to perform 3D spatial reasoning test by using large amounts of the unlabeled training data and considerably small set of labelled examples.

**Evaluation.**   In order to evaluate the 3D reconstruction ability of the model we used Hausdorff distance (Taha & Hanbury, 2015) and MSE. Hausdorff distance measures the model's 3D reconstruction's correspondence with the input for a given camera pose. We measure the correctness of the recovered depth using standard MSE with respect to ground truth depth maps. We evaluate the 3D generation generation qualitatively. Finally the evaluation metric for the 3D-IQTT is the percentage of correctly answered *questions*.

**Datasets.**   We have created multiple different scene datasets ranging from simple to complex in nature. Those scenes are composed by a room containing one or more objects placed at random positions and orientations. Each 3D scene is rendered into a single $128 \times 128 \times 3$ image taken from a camera in a random sampled uniformly on the positive octant of a sphere containing the room. Technically, the probability of seeing the same configuration of a scene from two different views is near zero. *Box scenes* is created with simple box 3D shape (as depicted in Figure 14). *Shape scenes* is created with basic 3D shapes (i.e., box, sphere, cone, torus, teapot etc). *ShapeNet scenes* is composed by 6 objects from the ShapeNet dataset (Chang et al., 2015) (i.e., bowls, bottles, mugs, cans, caps and bags). For 3D-IQTT task we generated a test where each IQ question instance consists of a reference image of Tetris-like shape, as well as 3 other images, one of which is a randomly rotated version of the reference (see Figure 10 for an example). The training set is formed by 200k questions where

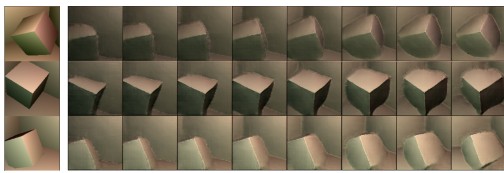 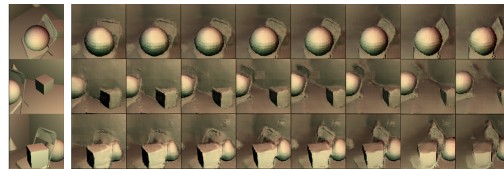

Figure 5: **View point reconstruction.** Given a scene (first column), we rotate the camera around to visualize the models understanding of 3D shape. As shown, the model correctly infers the unobserved geometry of the objects, demonstrating true 3D understanding of the scene. Videos of these reconstructions can be seen at https://bit.ly/2zADuqG.

| | Shape scenes | | | | Multiple-shape scenes | | | |
|---|---|---|---|---|---|---|---|---|
| | 5° | 35° | 55° | 80° | 5° | 35° | 55° | 80° |
| Hausdorff-F | 0.110 | 0.143 | 0.140 | 0.161 | 0.256 | 0.301 | 0.282 | 0.272 |
| Hausdorff-R | 0.156 | 0.191 | 0.189 | 0.202 | 0.308 | 0.355 | 0.329 | 0.316 |
| MSE-depth | 0.012 | 0.021 | 0.022 | 0.027 | 0.070 | 0.091 | 0.088 | 0.083 |

Table 2: **View point reconstruction.** Quantitative evaluation of implicit 3D reconstruction for unseen views by extrapolating the view angle from $0°$(original) to $80°$.

only a few are labelled with the information about the correct answer (i.e., either $5\%$ (10k) or $0.5\%$ (1k) of the total training data). The validation and test sets each contain 100K labelled questions. More details on experimental setup and evaluation can be found in appendix B.

## 4.2 IMPLICIT 3D SCENE RECONSTRUCTION

Figure 4 shows the input *Shape scenes* data and its corresponding reconstructions, along with its recovered depth and normal maps. The depth map is encoded in such a way that the darkest points are closer to the camera. The normal map colors correspond to the cardinal directions (red/green/blue for x/y/z axis respectively). Table 1 shows a quantitative evaluation of the forward and reverse Hausdorff distances on three different datasets. The table also depicts mean squared error of the generated depth map with respect to the input depth map. Figure 6 shows the reconstructions from the model on more challenging *multiple-shape scenes* where the number of objects as well as their shape varies. Figure 16 in the appendix showcases more qualitative evaluations.

To showcase that our model can reconstruct unobserved views, we infer the latent code $z$ of an image $x$ and then we decode and render different views while rotating the camera around the scene. Table 2 shows the Hausdorff distance and MSE of reconstructing a scene from different unobserved view angles. As the view angle increases from $0°$(original) to $80°$ for *shape scenes* the reconstruction Error and MSE tend to increase. However, for the *multiple-shape scenes* setup the trend is not as clear because of the complexity of the scene an the inter-object occlusions. Figure 5 qualitatively shows how pix2scene correctly infers the extents of the scene not in view in a consistent manner, demonstrating true 3D understanding of the scene.

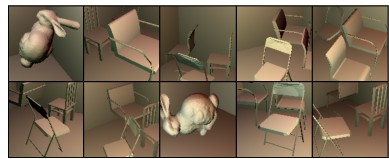 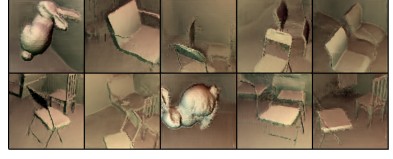

(a) Input images                              (b) Reconstructed images

Figure 6: **Multiple-shape scenes reconstruction.** Implicit 3D reconstruction of scenes composed by multiple ShapeNet objects.

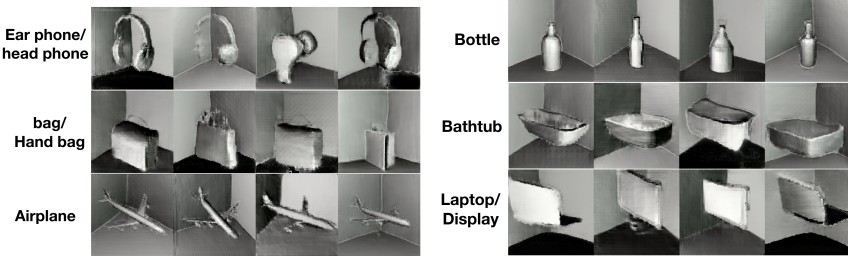

Figure 7: **Unconditional scene generation.** Generated samples from pix2scene model trained on ShapeNet scenes. **Left:** shaded images; **Right:** depth maps

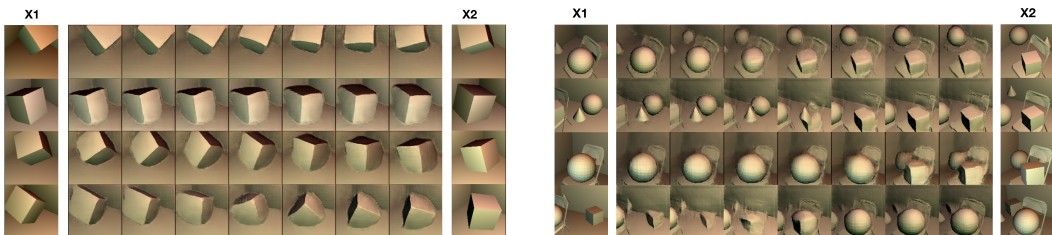

Figure 8: **Conditional scene generation.** Class conditioned generated samples for ShapeNet dataset.

### 4.3 IMPLICIT 3D SCENE GENERATION

We trained pix2scene on scenes composed of ShapeNet objects. Figure 7 shows qualitative results on unconditional generation. This shows how our model is able to generate correct 3D interpretations of the world. We also trained our model conditionally by giving the class label of the ShapeNet object to the decoder and critic networks (Mirza & Osindero, 2014). Figure 8 shows the results of conditioning the generator on different target classes.

In order to explore the manifold of the learned representations we select two images $x_1$ and $x_2$ from the held out data, then we linearly interpolate between their encodings $z_1$ and $z_2$ and decode the intermediary points into their corresponding images. Figure 9 shows this for two different settings. In each case, our representations capture the major geometrical aspects of the scene.

### 4.4 3D-IQ TEST TASK

We have designed a quantitative evaluation for 3D reconstruction which we refer to as the 3D-IQ test task (3D-IQTT). In their landmark work, Shepard & Metzler (1971) introduced the mental rotation task into the toolkit of cognitive assessment. The authors presented human subjects with reference images and answer images and the subjects had to quickly decide if the answer was either a 3D-rotated version or a mirrored version of the reference. The speed and accuracy with which people can solve this mental rotation task has since become a staple of IQ tests like the Woodcock-Johnson test (Woodcock et al., 2001). We took these as inspiration when designing a quantitative evaluation: we are using the same kind of 3D objects but instead of confronting our model with pairs of images

Figure 9: **Manifold exploration.** Exploration of the learned manifold of 3D representations. Generated interpolations (middle columns) between two images $x_1$ and $x_2$ (first and last columns).

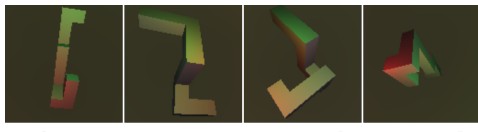 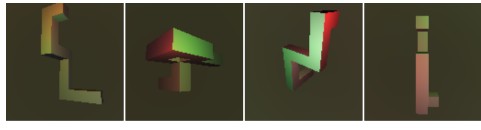

Figure 10: **Sample questions from the 3D-IQ test task.** For this "mental rotation" task, a set of reference images and 3 possible answers are presented. The goal is to find the rotated version of the reference 3D model. To solve this task, the human or the model has to infer the 3D shape of the reference from the 2D image and compare that to the inferred 3D shapes of the answers. The correct answers to these two examples are in the footnote.

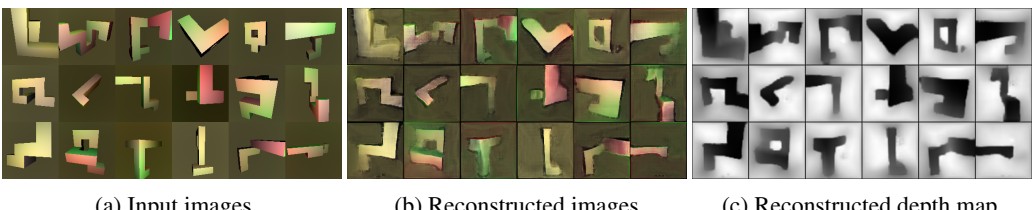

(a) Input images      (b) Reconstructed images      (c) Reconstructed depth map

Figure 11: **3D IQ test task.** Pix2scene reconstructions of the 3D-IQTT shapes.

and only two possible answers, we include several distractor answers and the subject (human or computer) has to to pick the correct answers out of 3 that is a 3D-rotated version of the reference object. To verify that our approach is able to learn accurate embeddings of these shapes we first assessed the reconstruction of these shapes qualitatively as shown in Figure 11.

### 4.4.1 SEMI-SUPERVISED CLASSIFICATION ON THE 3D-IQ TEST TASK

For training pix2scene in a semi-supervised setting, in addition to the unlabeled data, we also used the labelled data. The training with the unlabeled samples differs from the approach described for previous experiments as we do not assume that we have the knowledge of camera position. Thus, part of the latent vector $z$ encodes the actual 3D object (denoted as $z_{scene}$) and the remainder estimates the camera-pose (denoted as $z_{view}$). For the supervised training two additional loss terms were added: (a) a loss that enforces the object component ($z_{scene}$) to be the same for both the reference object and the correct answer, (b) a loss that maximizes the distance between object component of reference and the distractors. Losses (a) and (b) are contained in Equation 5 where $d_i$ denotes the distractors, $\boldsymbol{x}_{ref}$ is the reference and $\boldsymbol{x}_{ans}$ the correct answer. Algorithm is detailed in Table 1.

$$\mathcal{L}_\theta(\boldsymbol{x}_{ref}, \boldsymbol{x}_{d_1}, \boldsymbol{x}_{d_2}, \boldsymbol{x}_{ans}) = \frac{1}{2} D_\theta(\boldsymbol{x}_{ref}, \boldsymbol{x}_{ans}) - \frac{1}{2} \sum_{i=1}^{2} D_\theta(\boldsymbol{x}_{ref}, \boldsymbol{x}_{d_i}) \tag{5}$$

$$\text{where } D_\theta(\boldsymbol{x}_1, \boldsymbol{x}_2) = (||\boldsymbol{z}_{scene}^{\boldsymbol{x}_1} - \boldsymbol{z}_{scene}^{\boldsymbol{x}_2}||_2)^2 \text{ and } \boldsymbol{z}^{\boldsymbol{x}} = Encoder_\theta(\boldsymbol{x})$$

During the training we also minimize the mutual information between $z_{scene}$ and $z_{view}$ to explicitly disentangle and make sure that the learnt latent code has distinct source of information present in its dimensions. This is implemented via MINE (Belghazi et al., 2018). The strategy of MINE is to parameterize a variational formulation of the mutual information in terms of a neural network:

$$I_\Theta(z_s, z_v) = \sup_{\theta \in \Theta} \mathbb{E}_{\mathbb{P}_{z_s z_v}}[T_\theta] - \log(\mathbb{E}_{\mathbb{P}_{z_s} \otimes \mathbb{P}_{z_v}}[e^{T_\theta}]). \tag{6}$$

This objective is optimized in an adversarial paradigm where $T$, the statistics network, plays the role of the critic and is fed with samples from the joint and marginal distribution. We added this loss to our pix2scene objective to minimize the mutual information estimate in both unsupervised and

---

[1] three    [2] two

---

**Algorithm 1** Semisupervised classification

---

1: **while** $iter < max\_iter$ **do**
2:     $D \leftarrow \text{MiniBatch}()$
3:     $z \sim E(\boldsymbol{x}_{ref}); \forall (\boldsymbol{x}_{ref}, \boldsymbol{x}_{d_1}, \boldsymbol{x}_{d_2}, \boldsymbol{x}_{ans}) \in D$
4:     $L \leftarrow \mathcal{L}_{ALI} + \mathcal{L}_{recon} + I_{\Theta}(z_{scene}, z_{view})$
5:     **if** supervised-training-interval($iter$) **then**
6:         $L \leftarrow L + \mathcal{L}_{\theta}(\boldsymbol{x}_{ref}, \boldsymbol{x}_{d_1}, \boldsymbol{x}_{d_2}, \boldsymbol{x}_{ans})$
7:     **end if**
8:     optimize networks with $L$
9: **end while**

---

| Labeled Samples | CNN | Siamese CNN | Human Evaluation | Pix2Scene (Ours) |
|---|---|---|---|---|
| 0 (Unsupervised) | 0.3385 | 0.3698 | $0.7329 \pm 0.1488$ | $0.4372 \pm 0.0301$ |
| 200 | 0.3350 | 0.3610 | - | $0.4691 \pm 0.0259$ |
| 1,000 | 0.3392 | 0.3701 | - | $0.5567 \pm 0.0095$ |
| 10,000 | 0.3649 | 0.3752 | - | $0.5983 \pm 0.0021$ |

Table 3: **3D-IQTT quantitative results.** The test accuracy of the 3D-IQ test task show that the CNN baselines struggle to solve this task Pix2scene is able to understand the underlying 3D structure of the images and solve the task. The results show that although our model performs better than the baselines, we are still lagging behind the human level.

supervised training iterations. Once the model is trained, we answer 3D-IQTT questions, by inferring the latent 3D representation for each of the 4 images and we select the answer closest to the reference image as measured by L2 distance.

We compared our model to two different baselines. The first one is composed of 4 ResNet-50 modules (He et al., 2016) with shared weights followed by 3 fully-connected layers. We trained this CNN only on the labeled samples. Our second baseline has a similar architecture as the previous one but the fully-connected layers were removed. Instead of the supervised loss provided in the form of correct answers, it is trained on the contrastive loss (Koch et al., 2015). This loss reduces the feature distance between the references and correct answers and maximizes the feature distance between the references and incorrect answers. A more detailed description of the networks and contrastive loss function can be found in the appendix D.

Table 3 shows 3D-IQTT results for our method and baselines. The baselines were not able to interpret the underlying 3D structure of the data and its results are only slightly better than a random guess. The poor performance of the Siamese CNN might be in part because the contrastive loss rewards similarities in pixel space and has no notion of 3D similarity. However, pix2scene achieved significantly better accuracy by leveraging the learned 3D knowledge of objects.

## 5 CONCLUSIONS

In this paper we proposed a generative approach to learn 3D structural properties from single images in an unsupervised and implicit fashion. Our model receives an image of a scene with uniform material as input, estimates the depth of the scene points and then reconstructs the input scene. We also provided quantitative evidence that support our argument by introducing a novel IQ-task in a semi-supervised setup. We hope that this evaluation metric will be used as a standard benchmark to measure the 3D understanding capability of the models across different 3D representations. The main drawback of our current model is that it requires the knowledge of lighting and material properties. Future work will focus on tackling the more ambitious setting of learning complex materials and texture along with modelling the lighting properties of the scene.

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

## A  ARCHITECTURE

Pix2scene is composed of an encoder network(See Table 4), a decoder network(See Table 5), and a critic network(See Table 6). Specifically, the decoder architecture is similar to the generator in DCGAN (Radford et al., 2015) but with LeakyReLU (Mikolov et al., 2011) as activation function and batch-normalization (Ioffe & Szegedy, 2015). Also, we adjusted its depth and width to accommodate the high resolution images accordingly. In order to condition the camera position on the $z$ variable, we use conditional normalization in the alternate layers of the decoder. We train our model for 60K iterations with a batchsize of 6 with images of resolution $128 \times 128 \times 3$.

| Layer | Output size | Kernel size | Stride | BatchNorm | Activation |
|---|---|---|---|---|---|
| Input $[x, c]$ | $128 \times 128 \times 3$ | | | | |
| Convolution | $64 \times 64 \times 85$ | $4 \times 4$ | 2 | Yes | LeakyReLU |
| Convolution | $32 \times 32 \times 170$ | $4 \times 4$ | 2 | Yes | LeakyReLU |
| Convolution | $16 \times 16 \times 340$ | $4 \times 4$ | 2 | Yes | LeakyReLU |
| Convolution | $8 \times 8 \times 680$ | $4 \times 4$ | 2 | Yes | LeakyReLU |
| Convolution | $4 \times 4 \times 1360$ | $4 \times 4$ | 2 | No | LeakyReLU |
| Convolution | $1 \times 1 \times 1$ | $4 \times 4$ | 1 | No | |

Table 4: **Pix2scene encoder architecture**

| Layer | Output size | Kernel size | Stride | BatchNorm | Activation |
|---|---|---|---|---|---|
| Input $[x, c]$ | $131 \times 1$ | | | | |
| Convolution | $4 \times 4 \times 1344$ | $4 \times 4$ | 1 | Yes | LeakyReLU |
| Convolution | $8 \times 8 \times 627$ | $4 \times 4$ | 2 | Yes | LeakyReLU |
| Convolution | $16 \times 16 \times 336$ | $4 \times 4$ | 2 | Yes | LeakyReLU |
| Convolution | $32 \times 32 \times 168$ | $4 \times 4$ | 2 | Yes | LeakyReLU |
| Convolution | $64 \times 64 \times 84$ | $4 \times 4$ | 2 | Yes | LeakyReLU |
| Convolution | $128 \times 128 \times nCh$ | $4 \times 4$ | 2 | Yes | |

Table 5: **Pix2scene decoder architecture.**

| Layer | Output size | Kernel size | Stride | BatchNorm | Activation |
|---|---|---|---|---|---|
| Input $[x, c]$ | $128 \times 128 \times 6$ | | | | |
| Convolution | $64 \times 64 \times 85$ | $4 \times 4$ | 2 | No | LeakyReLU |
| Convolution | $32 \times 32 \times 170$ | $4 \times 4$ | 2 | No | LeakyReLU |
| Convolution | $16 \times 16 \times 340$ | $4 \times 4$ | 2 | No | LeakyReLU |
| Convolution | $8 \times 8 \times 680$ | $4 \times 4$ | 2 | No | LeakyReLU |
| Convolution + [z] | $4 \times 4 \times 1360$ | $4 \times 4$ | 2 | No | LeakyReLU |
| Convolution | $1 \times 1 \times 1$ | $4 \times 4$ | 1 | No | |

Table 6: **Pix2scene critic architecture.** Conditional version takes image, latent code $z$ and camera position $c$.

## B  MATERIAL, LIGHTS, AND CAMERA PROPERTIES

**Material.**  In our experiments, we use diffuse materials with uniform reflectance. The reflectance values are chosen arbitrarily and we use the same material properties for both the input and the generator side.

**Camera.**  The camera is specified by its position, viewing direction and vector indicating the orientation of the camera. The camera positions were uniform randomly sampled on a sphere for the 3D-IQTT task and on a spherical patch contained in the positive octant, for the rest of the experiments. The viewing direction was updated based on the camera position and the center of mass of the objects, so that the camera was always looking at a fixed point in the scene as its position changed. The focal

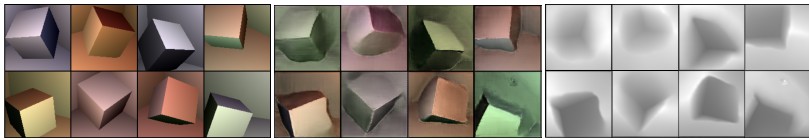

    (a) Input images     (b) Reconstructed images  (c) Reconstructed depth map

Figure 12: **Random lights configuration.**

length ranged between [18 mm and 25mm] in all the experiments. and field-of-view was fixed to 24mm . The camera properties were also shared between the input and the generator side. However, in the 3D-IQTT task we relax the assumption that we know the camera pose and instead estimate the view as a part of the learnt latent representation.

**Lights.** For the light sources, we experimented with single and multiple point-light sources, with the light colors chosen randomly. The light positions are uniformly sampled on a sphere for the 3D IQTT tasks, and uniformly on a spherical patch covering the positive octant for the other scenes. The same light colors and positions are used both for rendering the input and the generated images. The lights acted as a physical spot lights with the radiant energy attenuating quadratically with distance. As an ablation study we relaxed this assumption of having perfect knowledge of lights by using random position and random color lights. Those experiments show that the light information is not needed by our model to learn the 3D structure of the data. However, as we use random lights on the generator side, the shading of the reconstructions is in different color than in the input as shown in Figure 12.

## C EVALUATION OF 3D RECONSTRUCTIONS

For evaluating 3D reconstructions, we use the Hausdorff distance (Taha & Hanbury, 2015) as a measure of similarity between two shapes as in Niu et al. (2018). Given two point sets, $A$ and $B$, the Hausdorff distance is, $\max \left\{ \max D_H^+(A, B), \max D_H^+(B, A) \right\}$, where $D_H^+$ is an asymmetric Hausdorff distance between two point sets. E.g., $\max D_H^+(A, B) = \max D(a, B)$, for all $a \in A$, or the largest Euclidean distance $D(\cdot)$, from a set of points in $A$ to $B$, and a similar definition for the reverse case $\max D_H^+(B, A)$.

## D ARCHITECTURE FOR 3D IQTT EVALUATIONS

Pixel2Scene architecture remains similar to the ones in previous sections but with higher capacity on decoder and critic as this task is more challenging and complex. The more important difference is that for those experiments we do not condition the networks with the camera pose to be fair with the baselines. In addition to the three networks we have a statistics network (see Table 7) that estimates and minimizes the mutual information between the two set of dimensions in the latent code using MINE (Belghazi et al., 2018). Out of 128 dimensions for $z$ we use first 118 dimensions for represent scene-based information and rest to encode view based info.

The architecture of the baseline networks is shown in Figure 13. The contrastive loss using for training this baselines is shown in Figure 7.

| Layer | Output size | Kernel size | Stride | BatchNorm | Activation |
|---|---|---|---|---|---|
| Input $[z[:118], z[118:]]$ | $1 \times 1 \times 128$ | | | | |
| Convolution | $1 \times 1 \times 256$ | $1 \times 1$ | 1 | No | ELU |
| Convolution | $1 \times 1 \times 512$ | $1 \times 1$ | 1 | No | ELU |
| Convolution | $1 \times 1 \times 1$ | $1 \times 1$ | 2 | No | None |

Table 7: **Pix2scene statistics network architecture.**

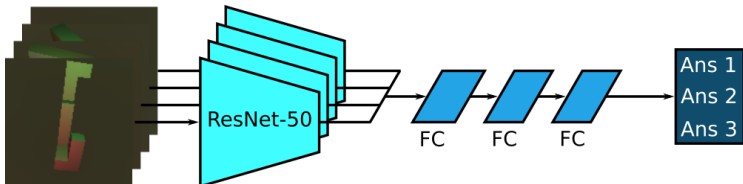

Figure 13: **3D-IQTT baseline architecture.** The ResNet-50 all share the same weights and were slightly modified to support our image size. "FC" stands for fully-connected layer and the hidden node sizes are 2048, 512, and 256 respectively. The output of the network is encoded as one-hot vector.

The contrastive loss from Equation 7 is applied to the 2048 features that are generated by each ResNet block. $\boldsymbol{x}_1$ and $\boldsymbol{x}_2$ are the input images, $y$ is either 0 (if the inputs are supposed to be the same) or 1 (if the images are supposed to be different), $G_\theta$ is each ResNet block, parameterized by $\theta$, and $m$ is the margin, which we set to 2.0. The loss function is from Hadsell et al. (2006) but used slightly differently.

$$\mathcal{L}_\theta(\boldsymbol{x}_1, \boldsymbol{x}_2, y) = (1 - y)\frac{1}{2}(D_\theta(\boldsymbol{x}_1, \boldsymbol{x}_2))^2 + (y)\frac{1}{2}(max(0, m - D_\theta(\boldsymbol{x}_1, \boldsymbol{x}_2)))^2$$
$$D_\theta(\boldsymbol{x}_1, \boldsymbol{x}_2) = ||G_\theta(\boldsymbol{x}_1) - G_\theta(\boldsymbol{x}_2)||_2 \tag{7}$$

## E  MORE SCENE RECONSTRUCTIONS

Figure 14 shows 3D reconstructions of scenes formed by boxes in a room. In Figure 15 our model is asked to reconstruct the scenes of the first column and then render different views of the same scene. In this case we show the normal maps of those views. Figure 16 shows the recovered shading, depth and normal images from reconstructions of complex scenes such as bedrooms and bunny.

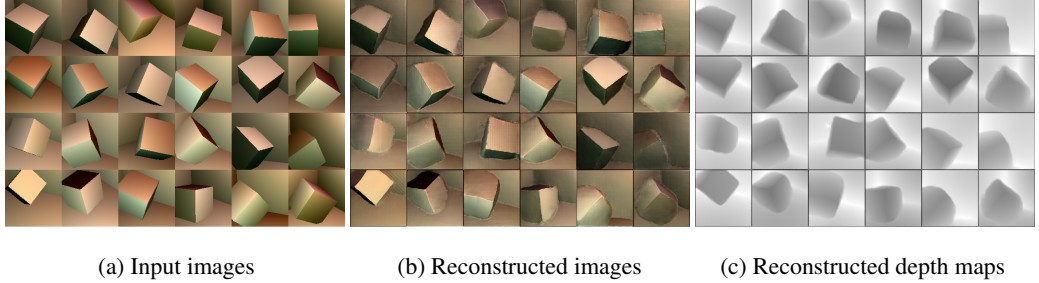

(a) Input images          (b) Reconstructed images          (c) Reconstructed depth maps

Figure 14: **Scene reconstruction.** (a) Input images of rotated cubes into a room. (b) pix2scene reconstructions with its (c) associated depth maps.

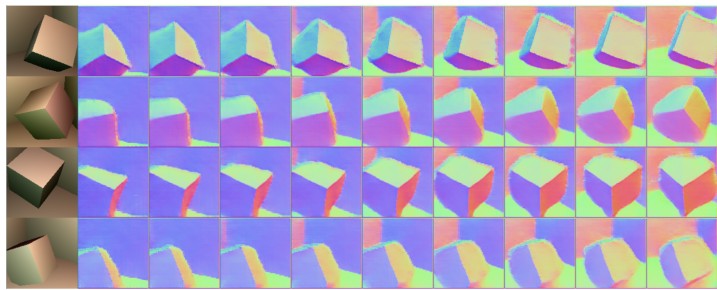

Figure 15: **Normal views reconstruction.** For each row, the first column is the input image and other columns are the extrapolated normal maps of that image from different views.

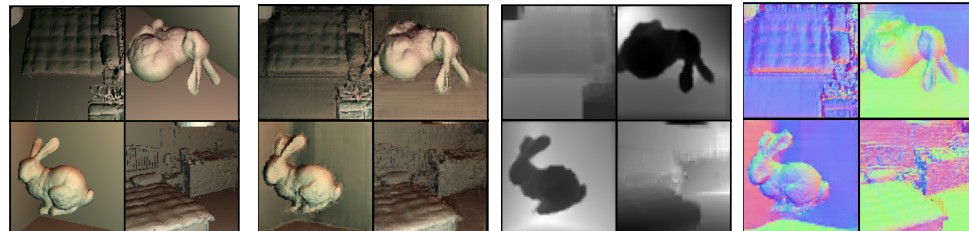

(a) Input images    (b) Reconstructed images (c) Reconstructed depth (d) Reconstructed normal

Figure 16: **Reconstruction of complex scenes.** Reconstruction of bedroom scenes and bunny.

