# OpenReview forum: "Pix2Scene: Learning Implicit 3D Representations from Images"
_ICLR.cc/2019/Conference_

### Official Review · AnonReviewer2 · 2018-11-02
**learning 3D or depth images from 2D images**

**Rating:** 6
**Confidence:** 1

**Review:**

The paper deals with creating 3D representations or depth maps from 2D image data using adversarial training methods.
The flow makes the paper readable.

One main concern is that most of the experiments seem to have results as visual inspections of figures provided. It is really hard to judge the correctness or how well the algorithms do.

It would be useful to provide references of equation 1 if used from previous text.

In the experiments, it is usually not clear how many training images were used, how many test. How different were the objects used in the training data vs test? Were all the test objects novel? How useful were the GAN techniques? Which part of the GAN did the most work i.e. the usefulness and accuracy of the different parts of the net? Even in 4.2, though it mentions use of 6 object types for both training and testing, using the figures is hard to estimate how well the model does compared to a reference baseline.

In 4.4.1, the discussion on how much improvement there is due to use of unlabeled images is missing? Do they even help? It is not quite clear from table 1. How many unlabeled images were used? How many iterations in total are used of the unlabeled ones (given there is 1 in 100 update of labeled ones).

Missing reference: http://www.cs.cornell.edu/~asaxena/reconstruction3d/saxena_make3d_learning3dstructure.pdf

---

> ### Author Response · Authors · 2018-11-18
> **Reply to Reviewer 2**
>
> Dear reviewer, thank you for your time and effort.
> TL;DR: We added more quantitative results. Our paper already includes examples generalizing to viewpoints that weren’t part of the training data, but we included additional samples. And we added significantly more details about the methods.
>
> Here are our responses in more detail:
>
>
> == RE: Most evaluations are qualitative.
>
> There is no standard protocol to evaluate 3D reconstruction and generation. Most of the state-of-the-art methods (fully unsupervised methods learned on single images) just show qualitative results in their papers.
> We did quantitatively evaluate the surfel reconstruction against the ground truth via Hausdorff distance (HD) as described in Appendix B and the reconstructions of our model via mean squared error (MSE) on the depth map. We achieved near-zero MSE and reasonably lower HD (when reconstructed from the same view). We included a table showing the difference in these values for different, unseen camera views: https://ibb.co/nj73qL. On top of these metrics, we also created the 3D IQ test task (3D-IQTT) which is exclusively quantitative. We compared our method with two CNN baselines and we now also included human evaluation. The CNN baselines demonstrate that the task can only be solved with an understanding of the 3D geometry. A preview of the updated comparison table can be found here: ( https://ibb.co/nhHUS0) .
>
>
> == RE: Add reference for the rendering equation.
>
> Sorry for the oversight. We have added the reference for the rendering equation: it is an approximation of Kajiya’s rendering equation [Kajiya 1986].
>
> [Kajiya 1986] “The Rendering Equation”
>
>
> == RE: More details on experimental setup.
>
> We have added more details on the experimental setup (camera, lights, and material properties used) in the appendix. All our images are of resolutions 128x128.
> Except for the 3D-IQTT, we didn’t store a fixed dataset but rather created the dataset on the fly. For example in the existing Figure 4, during the data generation process the rotation, translation, and object categories were randomized. The probability of seeing the same configuration from two different views is near zero.
>
>
> == RE: Which parts of the GAN are more important.
>
> Our Pix2Scene architecture is a bidirectional adversarial model. It consists of an encoder, decoder, renderer, and discriminator. The encoder translates the input image into a latent representation. The decoder transforms a similar latent vector, sampled from noise, into our surfel representation, which is converted into a 2D image by the renderer. The discriminator’s purpose is to make sure the output images become the same distribution as the input images, and ascertain that the encoded latent representation corresponds to the latent input to the decoder. See our existing Figure 3 for an overview. The decoder-rendering part is important for generating new viewpoints for a given latent code and the encoder-decoder pipeline allows us to infer the 3D structure of a 2D image. Without the encoder, the model would be purely generative.
>
>
> == RE: Novelty of the generated images.
>
> GAN-based models usually suffer from mode-collapse. We demonstrated in Figure 8 that our model overcame this issue and was able to interpolate between two given scenes. We’ve added another figure to further emphasize the interpolation capabilities of our model.
>
>
> == RE: 3D-IQTT semisupervised learning.
>
> Thanks for this feedback. We agree that this section wasn’t sufficiently clear. We’ve rewritten a part of this section and added the details on the interleaved training. It's similar to algorithm 2 from [Kingma et al. 2014], except instead of a randomized minibatch, we train a few iterations of unsupervised data followed by a few iterations of supervised data. We also extended Table 1 to include an entirely unsupervised case as well as human performance on the same task. A preview of the table can be found here: https://ibb.co/nhHUS0. In all cases, the model was trained with an unsupervised dataset of 100,000 lines of data, where each line contained the reference image, the 3 possible answers, but no information on which one was the correct answer.
>
> [Kingma et al. 2014] "Semi-supervised Learning with Deep Generative Models", 2014.
>
>
> == RE: Missing reference Make-3D.
>
> Sorry for the oversight. We will add the reference in our introduction.

---

### Official Review · AnonReviewer1 · 2018-11-02
**Nice model but some details missing**

**Rating:** 6
**Confidence:** 4

**Review:**

This paper introduces a method to create a 3D scene model given a 2D image and a camera pose. The method is: (1) an "encoder" network maps the image to some latent code vector, (2) a "decoder" network uses the code and the camera pose to create a depthmap, (3) surface normals are computed from the depthmap, and (4) these outputs are fed to a differentiable renderer which reconstructs the input image. At training time, a discriminator provides feedback to (and simultaneously trains on) the latent code and the reconstructions. The model is self-supervised by the reconstruction error and the GAN setup. Experiments show compelling results in 3D scene generation for simple monochromatic synthetic scenes composed of an empty room corner and floating ShapeNet shapes.

This is a nice problem, and if the approach ever works in the real world, it will be useful. On synthetic environments, the results are impressive.

The paper seems to claim more ground than it actually covers. The abstract says "Our method learns the depth and orientation of scene points visible in images", but really only the depth is learned, and the "orientation" is an automatically-computed surface normal, which is a free byproduct of any depth estimate. The "surfel" description includes a reflectance vector, but this is never estimated or further described in the paper, so my guess is that it is simply treated as a scalar (which equals 1). Taking this reflectance issue together with the orientation issue, the model is not really estimating surfels at all, but rather just a depthmap, which makes the method seem considerably less novel. Furthermore, the differentiable rendering (eq. 1) appears to assume that all light sources are known exactly -- this is not a trivial assumption, and yet it is never mentioned in the paper. The text suggests that only an image is required to run the model, but Figure 3 shows that the networks are conditioned on the camera pose -- exact knowledge of the camera pose is difficult to obtain precisely in real settings, so this again is not an assumption to ignore.

To rewrite the paper more plainly, one might say that it receives a monochrome image as input, estimates a depthmap, and then shades this depthmap using perfect knowledge of lighting and camera pose, which reconstructs the input. This may sound less appealing, but it also seems more accurate.

The paper is also missing some details of the method and evaluation, which I hope can be cleared up easily.
- What is happening with the light source? This is critical in the shading equation (eq. 1), and yet no information is given on it -- we need the color and the position of every light in the scene.
- How is the camera pose represented? Section 3.3.3 says conditional normalization is used, but what exactly is fed to the network that estimates these conditional normalization parameters?
- What is the exact form of the reconstruction error? An equation would be great.
- How is the class-conditioning done in 4.2?
- In Eq. 4, the first usage of D_\theta should use only the object part of the vectors, and the second usage should use only the geometric part, right? Maybe this can be cleared up with a second D_subscript.
- I do not understand the "interleaved" training setup in 4.4.1. Please explain that more.
- It is not clear to me why the task in 4.4.2 needs any supervised training at all, if the classification is just done by computing L2 distances in the latent space. What happens with "0 sampled labels"?

Overall, I like the paper, and I can imagine others in my group liking it. I hope it gets in, assuming the technical details get cleaned up and the language gets softer.

---

> ### Author Response · Authors · 2018-11-17
> **Reply to Reviewer 1's "Nice model but some details missing" (part 1 of 2).**
>
> Thanks so much for your time.
> TL;DR: We’re completely rewriting the intro to focus on our actual contribution and not the long-term plan. We’re also working towards removing all the assumptions like camera position and light position knowledge. And we’ve added a lot more details about the 3D-IQTT methods.
>
> == RE: Providing clarification on the claims.
>
> We agree that the extent of the contributions claimed in the introduction was unclear. We set out to present our long-term goal, to “learn the 3D structure of the real world just from single images”; however, in this paper, we have just made the first steps towards the goal and that was not apparent from the original introduction. We have reworked our claims accordingly in our new paper version (will be uploaded in the next few days) to reflect that our method “receives a monochrome image as input, estimates a depth map, and then shades this depth map using perfect knowledge of lighting and camera pose, which reconstructs the input” - as you suggested.
>
> Our model makes several assumptions: (a) the camera pose is known, (b) the material properties are constant, (c) the light positions are known, and (d) the world is piece-wise smooth. In order to achieve our long-term goal, we have to eventually get rid of these. Therefore we have made some first steps to address each one:
>
> - (Camera pose is known): In the 3D-IQTT experiments, we estimated the camera position in our latent representation while we kept the camera looking at the center of the object. We used the estimated camera parameters in the generator for the rendering process.
> - (Material properties are constant): We used diffuse materials with uniform reflectance for all our experiments. The reflectance values were chosen arbitrarily but kept fixed for input-output pairs. In other words, we use fixed material which can be chromatic (reflects different wavelengths by different amount) or monochromatic (reflects all wavelengths the same amount). This is not the same as using "monochromatic image", it is just that material is constant and doesn't need to be inferred.  We've added the details to the appendix.
> Learning the reflectance and color/texture properties (in addition to the surface depth and orientation) is significantly more challenging, but we are currently working towards that.
> - (Lighting assumptions):  In our work presented in the last version of the paper, we used multiple point light sources that were placed randomly on the surface of a spherical sector around the scene and colored randomly. For each pair of rendered input image and model-reconstructed output image, these light conditions were identical. We added more details about how we handled the lighting to the appendix.
> - (The world is piece-wise smooth): This might not be perfectly accurate, but it’s a common assumption in 3D reconstruction. In an extreme case like when capturing cactus spikes, this might not work, but for example, when we were reconstructing a chair with a thin stretcher (see our video, https://bit.ly/2zADuqG), the reconstruction worked well.
>
>
> We agree that we are currently mainly recovering a depth map, but the surfel representation was picked with our long-term goal in mind, since gives us several advantages: (a) surfel representation allow us to represent only the visible surface of a complicated scene instead of explicitly representing the complete scene. Given an image we can infer its implicit 3D representation and then recreate novel surfel representations of the underlying scene from unobserved viewpoints. Moreover this representation fits well with current convolutional architectures (b) with our existing normal estimation and additional material estimation this allows for realistic shading.

---

> ### Author Response · Authors · 2018-11-17
> **Reply to Reviewer 1's "Nice model but some details missing" (part 2 of 2).**
>
>
> == RE: How camera and class conditioning is done.
>
> For the view conditioning, we used conditional batch-normalization which transforms a 3-dimensional vector (representing the camera coordinates) into affine batch-normalization parameters. For the class conditioning, we used the standard conditional GAN technique: We encoded the class labels as one-hot vectors and concatenated this vector to the inputs of the decoder, encoder, and discriminator. We added all of this to the paper.
>
>
> == RE: Reconstruction error.
>
> We added the exact formula for the reconstruction loss to the paper.
> It is a weighted sum of two terms: (a) the L2 loss of the input image given to the encoder and the rendered output on the decoder, and (b) the L2 loss of the noise given to the decoder and the inferred latent code of the rendered decoder output. This loss is similar to the bi-directional L2-loss used in [Li et al. 2017] and [Huang et al. 2018].
>
> [Huang et al. 2018] “Multimodal Unsupervised Image-to-Image Translation”, 2018
> [Li et al. 2017] “ALICE: Towards Understanding Adversarial Learning for Joint Distribution Matching”, 2017
>
>
> == RE: 3D-IQTT loss function and training algorithm.
>
> In these experiments, we removed the assumption of knowing the camera position. Our model had to learn to represent the scene geometry in a part of the latent vector (z_scene) and the camera position in another part (z_view).
>
> The loss term for the supervised part of the training enforced this: it rewards the z_scene of the correct answer to be close to the z_scene of the reference and it pushes z_scene of the reference and z_scene of wrong answers apart. We also minimized mutual information between z_scene and z_view in order to enforce distinct source of information captured by the latent dimensions. We also improved the explanation of this method in the paper.
>
> We also added experiments where we did not add any supervised samples. In this case, z_scene and z_view get entangled and the task becomes significantly harder. Both CNN baselines were trained on the supervised data; therefore when comparing them in the unsupervised condition they perform according to the random initialization while our model was able to at least leverage the unsupervised data. Please see our updated Table-1 (https://ibb.co/nhHUS0 )
>
> We also added the interleaved training algorithm for the semi-supervised task to the paper. It's similar to algorithm 2 from [Kingma et al. 2014], except instead of a randomized minibatch, we train a few iterations of unsupervised data followed by a few iterations of supervised data.
>
> [Kingma et al. 2014] "Semi-supervised Learning with Deep Generative Models", 2014.

---

### Official Review · AnonReviewer3 · 2018-11-03
**Interesting inverse graphics model. Motivation and experiments are lacking.**

**Rating:** 5
**Confidence:** 4

**Review:**

This paper explored explaining scenes with surfels in a neural recognition model. The authors demonstrated results on image reconstruction, synthesis, and mental shape rotation.

The paper has many strengths. The model is clearly presented, the implementation is neat, the results on synthetic images are good. In particular, the results on the mental rotation task are interesting and new; I feel we should include more studies like these for scene and object representation learning.

A few concerns remain. First, the motivation of the paper is unclear. The main advantage of the proposed representation, according to the intro, is its `implicitness’, which enables viewpoint extrapolation. I’d like to see more explanation on why ‘explicit’ representations don’t support that. A lot of the intro is currently talking about related work, which can be moved to later sections or to the supp material.

The paper then moves on to discuss surfels. While it’s new combine surfels with deep nets, I’m not sure how much benefits it brings over voxels, point clouds, or primitives. It’d be good to compare with these scene representations.

My second concern is the results are all on synthetic data, and most shapes are very simple. While the paper is called ‘pix2scene’, it’s really about ‘pix2object’ or ‘pix2shape’. I’d like to see results on more realistic scenes, where the number of objects as well as their shape and material varies.

For the mental rotation task, the authors should cite and discuss the classic work from Shepard and Metzler and include human performance for calibration.

I’m on the border for this paper. Happy to adjust my rating based on the discussion and revision.

---

> ### Author Response · Authors · 2018-11-15
> **Quick question regarding comparison to voxel/point clouds/primitives**
>
> Dear AnonReviewer3,
>
> Thank you so much for your review. We appreciate the time you put into this and your feedback.
>
> Before we respond in full to all of your points, we have a quick question:
> You mentioned, "It’d be good to compare with these [voxel/point clouds/primitives] scene representations". We'd love to implement this but we're having some issues finding suitable code.  Do you know of any code for methods that implicitly (i.e. without supervised training) learns object reconstruction using meshes/voxels/point clouds?
>
> Thanks,
> the Pix2Scene team

---

> > ### Comment · AnonReviewer3 · 2018-11-15
> > **Response**
> >
> > Thanks for asking. I realize 'implicit' means without supervision.  In this case, how would the system compare with Rezende et al. [2016]?  Their code might not be available. An alternative is perspective transformer net [Yan et al, 2016] and its follow-ups.
> >
> > A common problem with these unsupervised/self-supervised (or 'implicit') approach is that the learned scene representation is often incomplete and looks bad from a different view. How would the reconstructed scenes (objects) in Fig 4 look like from a different view? These results are important for a '3D' representation.
> >
> > Rezende et al. Unsupervised Learning of 3D Structure from Images. NIPS'16.

---

> ### Author Response · Authors · 2018-11-22
> **Reply to Reviewer 3's "Interesting inverse graphics model" (part 1 of 2).**
>
> Thanks for your time and continued feedback
>
> TL;DR: We reworked the intro from the ground, drawing a much clearer distinction between our method and related methods. We also added the missing citation and conducted some human evaluations for our mental rotation task.
>
>
> == RE: The motivation of the paper is unclear.
>
> We have re-written the introduction to more clearly reflect the extent of our contributions. This revision will be released shortly.
>
>
> == RE: Advantages of Implicit representation over-explicit.
>
> Implicit and explicit 3D representations aren’t strictly better or worse but have significant advantages and disadvantages.
> An explicit representation stores all rendering-relevant information about all entities in a given 3D space. The main benefit of this method is that it’s easily transferable, in the sense that an explicit model can directly be loaded into a 3D modeling software and viewed from all angles without any inconsistencies. In the worst possible case, one would store all the properties of each voxel in 3D space, like opaqueness, illumination, color, reflectance, etc. For example, in a space of 512x512x512 points with 10 values per point which are float32-encoded, this would amount to 5GB of data for a single scene. The vast majority of these points aren’t relevant to most viewpoints, like the inside of objects. Therefore the common workaround is to use a sparse representation, like meshes. These do however come with their own drawbacks, like the question of how to discretize complex objects. This makes mesh representation difficult to deal with in neural networks. The current state of the art relies on deforming a pre-existing mesh.
> On the other hand, the implicit approach represents the 3D scene in a high-dimensional latent variable. The main drawback of this is the lack of interpretability: this vector on its own is meaningless and needs to be decoded into a viewpoint-dependent representation that can be rendered. Increasing a single value in this vector could, for example, illuminate an object more or morph a chair in the scene into a table, or both. The big benefit of this is scalability. This produces a compressed representation that fits the complexity of a given scene. The only remaining issue is the viewpoint-dependent consistency, i.e. making sure that when the scene is viewpoint-dependently decoded into a renderable model, the scene’s content is the same from all angles.
> We believe that neural networks can approximate this consistency-enforcing function and we think that in the long term when going to arbitrarily complex scenes, the advantages of this implicit representation outweigh the downsides.
>
> To summarize, in the explicit approach dealing with sparsity and discrete representations remains a major challenge for deep learning, and the cubic scaling of dense approaches like voxel-based representations makes them impractical for large scenes. On the other hand, our implicit representation generates plausible 3D models that show viewpoint extrapolation. This can directly be observed in our video ( https://bit.ly/2zADuqG ) for example in the rotating chair video, where we only fed the model a single chair image, and then move the camera to generate new viewing angles.
>
>
> == RE: Compare surfels with voxels, meshes and point clouds==
>
> Both meshes and voxels don’t scale well to scenes with multiple objects - voxels due to their poor scaling in complexity and meshes due to difficulties representing them with neural networks, as mentioned above. Point clouds don’t provide any surfaces. Therefore no shadows, no reflectance, and no shading are possible. In terms of evaluation, most contemporary works use supervised training for their voxel/mesh-reconstruction where we use unsupervised/self-supervised training. A direct comparison with these would be unfair. The only other method we found with unsupervised reconstructions was [Rezende et al. 2016] and there is no source code available for their method.
> We did include a new benchmark in the paper, the 3D-IQTT, that should allow for more methods to compare their reconstruction accuracy independently of the underlying method.
>
> [Rezende et al. 2016]  Unsupervised Learning of 3D Structure from Images.

---

> ### Author Response · Authors · 2018-11-22
> **Reply to Reviewer 3's "Interesting inverse graphics model" (part 2 of 2).**
>
>
> == RE: More realistic scenes and colors.
>
> Most previous approaches have focused on the reconstruction or generation of single objects with no background. In this work, we consider scenes formed by one or more objects in a simple room. We have added new results of more complicated scenes where the number of objects, as well as their shape, vary. Please see our new qualitative results ( https://ibb.co/cbL6AL ) and quantitative results ( https://ibb.co/nj73qL , right side of the table). Additional scene reconstructions can be found here: https://ibb.co/hjpBc0 . We included these new results in the paper, which we’ll update soon. We use diffuse materials with uniform reflectance for all our experiments. Learning the reflectance and color/texture properties (in addition to the surface depth and orientation) is significantly more challenging, but we are currently working towards that.
> We'll keep more realistic settings involving textures, shadows, and realistic lighting for future work.
>
>
> == RE: Missing citations in 3D-IQTT and human performance.
>
> For the 3D-IQTT we’ve carried out a human evaluation on over 40 students in the department, the results of which were included in the paper and can be seen at https://ibb.co/nhHUS0 . The results show that although our model performs better than the baselines, we are still lagging behind the human level. We also cited and discussed Shepard and Metzler's work.
>
>
> == RE: Move related work to the supplementary material.
>
> We completely rewrote the introduction to more clearly introduce the problem, challenges, assumptions, and the proposed method. We also moved unnecessary related work to the dedicated section.
>
>
> == RE: Comparison to “Unsupervised Learning of 3D Structure from Images” [Rezende et al. 2016] and “Perspective Transformer Nets” [Xinchen et al. 2016]
>
> Perspective Transformer Nets can be considered weakly supervised. They render 24 images per object, which are all observed by the networks during training. In most contexts, one needs a 3D object in order to obtain 24 images of an object. By contrast, we learn on a single image per individual scene, i.e. no scene configuration is ever seen twice. Moreover, the metric used in this paper was intersection over union which works well for voxel-based representation. It's not clear how this evaluation tool can be extended for surfels. Figure 9 in our paper demonstrated our capacity to new, unseen viewing angles. In order to evaluate our model’s reconstruction of a scene from different novel viewpoints, we added MSE evaluations on the depth map and Hausdorff distance evaluation on the reconstructed surfels from different camera angles. Please see our modified table ( https://ibb.co/nj73qL ). In principle, we could compare our approach with the method of Rezende, but the information provided in the paper is not sufficient to accurately reconstruct their model.

---

### Author Response · Authors · 2018-11-27
**New Revision Added**

Dear reviewers,

Thanks again for all comments and suggestions. We've significantly rewritten major parts of the paper according to your input. We think your feedback has improved the quality of the paper. We specifically clarified the message of our paper, our contributions, and our methods.

Thanks again for your time and for giving the new revision another look.

---

> ### Comment · AnonReviewer1 · 2018-11-29
> **Clear improvement**
>
> I think the revised paper is a clear improvement over the original. The descriptions are more modest and more accurate, and the method is more clear than before. The added experiments are also helpful.
>
> One new complaint is: for the human evaluation on 3D-IQ, I think you need to specify how many participants there were, who they were, how they were recruited, and how long they had for each question, how many questions they answered, etc. Please just say everything, so that the mean and variance can be replicated.
>
> The new Fig1 is good.
>
> I think my current score of "above acceptance threshold" is still appropriate. I encourage anonreviewers 2 and 3 to boost their scores, to help the paper get in.

---

> > ### Author Response · Authors · 2018-11-30
> > **Addendum for human evaluation**
> >
> > Dear Reviewer 1,
> >
> > Thanks, that's really lovely to hear.
> >
> > As for the human evaluation: oh yeah, that's a clear oversight and should be in the appendix. To answer your questions:
> >
> > We posted the questionnaire to our lab-wide mailing list, where 41 participants followed the call. The questionnaire had 1 calibration question where, if answered incorrectly, we pointed out the correct answers. For all successive answers, we did not give any participant the correct answers and each participant had to answer all 20 questions to complete the quiz.
> >
> > We also asked participants for their age range, gender, education, and for comments. While many commented that the questions were hard, nobody gave us a clear reason to discard their response. All participants were at least high school graduates currently pursuing a Bachelor's degree. The majority of submissions (78%) were male, whereas the others were female or unspecified. Most of our participants (73.2%) were between 18 and 29 years old, the others between 30 and 39. The resulting test scores are normally distributed according to the Shapiro-Wilk test (p<0.05) and significantly different from random choice according to 1-sample Student's t test (p<0.01).
> >
> > P.S.: If you are curious, you can take the anonymized test as well: https://goo.gl/forms/QzVxjh9XOzhlhiIr2

---

> ### Comment · AnonReviewer3 · 2018-11-30
> **Still on the border**
>
> The revision is helpful and, as R1 suggested, is clearly better than the original version, especially in presentation.
>
> My major concern however remains: all results are on synthetic, simple scenes. In particular, these synthetic scenes don't have lighting, material, and texture variations, making them considerably easier than any types of real images. While the authors claimed prior work all used mesh supervision, this is not true for many papers that reconstruct 3D shapes using 2D supervision alone (e.g. Kanazawa 2018 as cited, which worked well on real data, reconstructing both shape and texture).
>
> I keep my original rating 5. I won't be too much against accepting this paper, if the AC decides to do so, but I cannot champion it.

---

> > ### Author Response · Authors · 2018-11-30
> > **Details about Kanazawa (2018)**
> >
> > Dear Reviewer 3,
> >
> > Thanks for your time again. We’d like to respond to your remark about other work. We don’t think the comparison to Kanazawa (2018) is fair. They don’t use ground truth mesh representation, but they have a very strong shape feedback from (a) the ground truth silhouette of the objects (b) the symmetry assumption (c) ground truth semantic object-specific keypoints. In more detail: They only experiment with birds and for each bird, they manually annotated the silhouette. Together with the mirror-symmetry assumption along one center plane, this covers most of the body’s geometry. But in addition to this, the training images were also annotated with 14 keypoints like the beak, legs, etc. and during training they minimize the distance between the 3D model keypoints and the annotations. These three factors in combination create a strong 3D ground truth signal.
> >
> > While this work is indeed impressive, we feel it’s inaccurate to call it “unsupervised”. The goal of our method is to infer structure purely from images. Without any kind of segmentation or manual annotation, purely unsupervised. There is only one other work that does something similar in Rezende (2016). But, as we outlined in the revised paper, (a) they only demonstrate this on perfectly illuminated single objects, whereas we do multiple objects with varying light conditions and (b) since they didn’t release any code, it’s hard for us and the community to reproduce their work.
> >
> > Yes, our scenes are simple, but to our best knowledge, we are the first to actually do this reconstruction from single unrelated images, not multiple views of identical scenes and without any auxiliary loss or training data.
> >
> >
> > Refs:
> >
> > (Kanazawa 2018) - Learning Category-Specific Mesh Reconstruction
> > from Image Collections, https://arxiv.org/pdf/1803.07549.pdf
> >
> > (Rezende 2016) - Unsupervised Learning of 3D Structure from Images, http://papers.nips.cc/paper/6600-unsupervised-learning-of-3d-structure-from-images.pdf

---

> > ### Author Response · Authors · 2018-12-07
> > **Encouraging further discussion**
> >
> > Dear Reviewer 3,
> >
> > We were wondering if you had any time to consider our response (the one below this post, titled "Details about Kanazawa (2018)"). If we forgot to address any of your concerns, please let us know.
> >
> > We would like to emphasize that the scope of this paper is the scene geometry reconstruction. Dealing with other scene properties like material and lights are part of our ongoing work. Our current model handles arbitrary geometric complexity as we demonstrated in the paper going from simple shape primitives to complex scenes (such as the ones in Figure 6 and 16).

---

> > > ### Comment · AnonReviewer3 · 2018-12-09
> > > **Unchanged**
> > >
> > > Thanks for the follow-up. I'm still not convinced. My main argument is the results are not strong, as the test scenes are very simple. I'd like to see any of the following:
> > >
> > > 1) results on real data, similar to those used by Kanazawa et al, or simpler than those but still reasonably realistic. I understand they used supervision, but only in 2D. If your model can achieve comparable (or slightly worse) results without supervision, that'd be a clear contribution.
> > >
> > > 2) clear explanation of why your model works better than Rezende et al. I understand the difficulty in reproducing their results without a released implementation, but as they have achieved good results without supervision more than two years ago, at least a clear explanation with some qualitative demonstration would be necessary.
> > >
> > > 3) You said "our scenes are simple, but to our best knowledge, we are the first to actually do this reconstruction from single unrelated images". This is not really true, as your 'single unrelated images' are often different viewpoints of the same object. In this case, how would you compare with GQN (Eslami et al, Neural scene representation and rendering), which is also 'implicit'? I don't want to bring up new related work at this moment, but I'm concerned that your recent comment is claiming too much.
> > >
> > > Eslami et al, Neural scene representation and rendering. Science, 2018.
> > >
> > > I think the concern on how the system works is very relevant and important, because your results are mainly on datasets of just a few objects, and they degrade noticeably on slightly more complicated scenes (Fig 16). With the current set of results, I don't see how this system can be practically useful.

---

> > > > ### Author Response · Authors · 2018-12-12
> > > > **Clarification regarding the concerns**
> > > >
> > > > Dear Reviewer 3, thank you again for your time and comments.
> > > >
> > > > We want to clarify what we interpret to be a fundamental misunderstanding regarding our contributions and those of prior art.
> > > >
> > > > Works like those of Yan et al., Eslami et al.  train on _multiple images_ of the same scene  and Kanazawa et al. and others use various forms of weak supervision. We, on the other hand, only present a _single view_ per unique scene configuration in an unsupervised manner.
> > > >
> > > > During training, we present our model with multiple views of similar (but never identical!) scenes. Even for scenes with simple objects, object placement and orientation is always different (i.e., randomly rotated and translated) per image. This is a fundamental difference, and significantly changes the problem landscape and difficulty.
> > > >
> > > > Most prior works you referred to consider multiple views of each individual scene. We, on the other, learn to infer 3D structure from single samples of unique 3D scenes. We do so in a _completely unsupervised_ fashion.
> > > >
> > > > From this, we are able to then synthesize, e.g., many novel views of a fixed scene configuration. We also demonstrate the ability to (smoothly) explore the space of possible scene configurations. We also introduce mental rotation task as a benchmark to evaluate 3D understanding based models irrespective of the underlying 3D representation used.
> > > >
> > > > We are happy to further discuss and contextualize our contribution with respect to these works you’ve raised across your replies during our review discussion, including in the case of techniques where reproducibility would be very challenging. Given our clarification, we feel the case for differentiation from prior art, as well as practicality and novelty, become clear.
> > > >
> > > > Moving forward we plan on scaling our method to datasets (like ImageNet) where, for every 3D scene, only a single image is available. Texture variation is one of the key directions of future development needed to make this jump, we feel.
> > > >
> > > > As the other two reviewers agree, we feel that the challenge of this particular problem (single-view, many scenes; purely unsupervised) and our results are promising and exciting for the community. Moreover, we hope the direction we present will open up avenues of future work towards robust, unsupervised scene understanding and reconstruction.
> > > >
> > > > Refs:
> > > >
> > > > Eslami et al, Neural scene representation and rendering. Science, 2018.
> > > > Kanazawa 2018 - Learning Category-Specific Mesh Reconstruction
> > > > Yan et al, perspective transformer nets , 2016

---

> > > > > ### Comment · AnonReviewer3 · 2018-12-12
> > > > > **Thanks. No misunderstanding here.**
> > > > >
> > > > > Thanks for the note. I see your point and I don't think there is any misunderstanding here. As I said: your 'single unrelated images' are often different viewpoints of the same object. Though the rotation and translation of the object vary, because the objects are so simple, and there are so many training images, it's very likely that two images are essentially different views of almost-the-same scene. Theoretically every scene is unique; in practice, different scenes might be too similar to tell perceptually, given the small degrees of freedom. In this case, the problem setup is then very similar to the related work I listed.

---

### Meta-Review · Area_Chair1 · 2018-12-10

**Confidence:** 4
**Recommendation:** Reject

**Metareview:**

This paper proposes an approach for learning to generate 3D views, using a surfel-based representation, trained entirely from 2D images.  After the discussion phase, reviewers rate the paper close to the acceptance threshold.

AnonReviewer3, who initially stated "My second concern is the results are all on synthetic data, and most shapes are very simple", remains concerned after the rebuttal, stating "all results are on synthetic, simple scenes. In particular, these synthetic scenes don't have lighting, material, and texture variations, making them considerably easier than any types of real images."

The AC agrees with the concerns raised by AnonReviewer3, and believes that more extensive experimentation, either on more complex synthetic scenes or on real images, is needed to back the claims of the paper.  Particularly relevant is the criticism that "While the paper is called ‘pix2scene’, it’s really about ‘pix2object’ or ‘pix2shape’."